# Causes of Failure of Open Innovation Practices in Small- and Medium-Sized Enterprises

Fernando Almeida 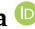

Polytechnic Higher Institute of Gaya (ISPGAYA) and INESC TEC, 4200-465 Porto, Portugal; almd@fe.up.pt

**Abstract:** The adoption of open innovation poses significant challenges that are important to explore. Studies in this field have mainly focused on exploring the causes of the failure of open innovation among large companies. This study addresses this research gap by employing a sample of 297 Portuguese small- and medium-sized enterprises (SMEs) to explore, through a quantitative study, whether the dimensions and causes of failure differ between large organizations and SMEs. A total of seven dimensions of causes of failure are considered, including strategy-related, organizational structure, organizational culture, knowledge and intellectual property management, management skill and action, resources, and interfirm collaboration. The findings reveal significant differences in four of these seven dimensions: the main causes of failure are related to the resources and management processes of open innovation in SMEs, while large companies face more challenges in the organizational structure and culture components. This study offers theoretical insights into the gaps in the literature to better understand the challenges facing open innovation. Furthermore, this study offers practical guidelines for SMEs to identify and mitigate these main obstacles, promoting better innovation management practices.

**Keywords:** open innovation; innovation management; failure; collaboration; SMEs

## 1. Introduction

The ability to innovate is increasingly crucial in today's rapidly evolving world. Various researchers, such as Ciocanel and Pavelescu (2015), Dempere et al. (2023), and Marto and Puertas (2023), have reported that innovation drives progress, competitiveness, and adaptability across various sectors including technology, business, healthcare, and education. In a constantly changing landscape, companies must innovate to stay ahead of the curve, meet customer demands, and solve emerging challenges. The ability to innovate allows businesses to differentiate themselves from competitors, creating unique value propositions and enhancing brand recognition. Beyond business, innovation plays a vital role in addressing societal issues such as climate change, healthcare disparities, and poverty (Fisher 2022; Guimarães et al. 2023).

The open innovation model arose as a response to the limitations of the traditional closed innovation approach, which relied solely on internal R&D activities to generate new ideas and bring products to the market. Recognizing these challenges, Chesbrough (2003) popularized the concept of open innovation in his seminal book *Open Innovation: The New Imperative for Creating and Profiting from Technology*. In their book, Chesbrough (2003) argues that this closed approach was becoming increasingly unsustainable in an era of accelerating technological change and global competition. Open innovation challenges the traditional boundaries of organizations by advocating for the inflow and outflow of ideas, knowledge, and resources between internal and external stakeholders. It emphasizes collaboration, partnerships, and cocreation with customers, suppliers, universities, and even competitors. By leveraging external sources of innovation, companies can access a broader pool of expertise, accelerate the pace of innovation, and reduce R&D costs (Sá et al. 2023).

The emergence of digital technologies and the internet facilitated the adoption of open innovation practices by providing platforms and tools for collaboration and knowledge sharing. One key aspect of digital technology's impact on open innovation is its ability to connect individuals and organizations across geographical boundaries (Urbinati et al. 2020). Through online platforms, companies can easily collaborate with external partners, including other businesses, research institutions, and even individual innovators. This interconnectedness fosters a rich ecosystem of knowledge exchange and cocreation, enabling organizations to tap into a diverse pool of expertise and ideas. Moreover, digital technologies have democratized the innovation process by lowering barriers to entry. Crowdsourcing platforms, for instance, allow companies to solicit ideas and solutions from a broad community, tapping into the collective intelligence of the crowd (Cricelli et al. 2022). Similarly, open-source software development has flourished, with developers worldwide collaborating on projects and freely sharing code. Furthermore, digital tools such as big data analytics and artificial intelligence enable organizations to extract insights from vast amounts of data, informing their innovation strategies and decision-making processes. By analyzing market trends, consumer behavior, and emerging technologies, researchers like Alghamdi and Agag (2023) and Capurro et al. (2022) have reported that companies can identify opportunities for innovation and adapt more quickly to changing market dynamics.

Open innovation practices have gained significant relevance in small- and medium-sized enterprises (SMEs) due to their potential to enhance competitiveness and foster growth in today's dynamic business environment (ACE 2012; Vanhaverbeke 2017). SMEs often face resource constraints, making it challenging to innovate internally (Bonanno et al. 2022; Istipliler et al. 2023). Open innovation allows them to tap into external expertise, ideas, and resources, leveraging collaboration with other firms, research institutions, and even customers. Open innovation is a double-edged sword, with a fine line between success and failure, as reported by Greco et al. (2022). Open innovation, while promising, comes with its share of failures, challenges, and risks. Several challenges and risks are identified in studies like Chaudhary et al. (2022) and Dabic et al. (2023), which report issues in managing intellectual property rights, loss of competitive advantage, leakage of sensitive information, and reputational damage in case of unsuccessful partnerships, among others. A systematic review of the literature on the causes of failure of open innovation was performed by Cricelli et al. (2023), who concluded that rigid structures and strict hierarchies prevent collaboration and the achievement of significant benefits in collaborative innovation processes. However, the research on open innovation failures remains relatively sparse, presenting a notable gap in the existing literature. Of particular note in this field are the studies carried out by Costa et al. (2023), who looked at the costs of engagement, and Bertello et al. (2022), who explored the challenges of university–industry–government collaboration. Consequently, while there is a growing body of research on open innovation and its benefits, the focus has often leaned toward successful cases or larger corporations. Understanding the factors contributing to failures in open innovation initiatives within SMEs is crucial for enhancing their innovation processes and overall competitiveness. Furthermore, Bertello et al. (2023) and Madanaguli et al. (2023) have highlighted this field as a relevant research agenda in the open innovation paradigm. This study responds to this challenge by conducting a quantitative analysis considering 297 Portuguese SMEs, identifying and exploring the extent of the shortcomings in the adoption of open innovation by SMEs.

The rest of this paper is organized as follows: First, a theoretical background of the topic is given, and the research hypotheses that guided this study are defined. This is followed by a presentation of the characteristics of the sample and the methods used to explore the results. Next, the main findings are presented, and their relevance to understanding the differences between SMEs and large organizations is discussed. Finally, the main conclusions are summarized, the theoretical and practical contributions are addressed, and suggestions for future work are provided.

## 2. Theoretical Background

Open innovation is a strategic approach that emphasizes the flow of ideas, knowledge, and resources both into and out of an organization. The concept of open innovation challenges the traditional closed model of innovation, which relies solely on internal research and development (R&D) efforts. Instead, it advocates for a more collaborative and inclusive approach to innovation, drawing on external sources such as customers, suppliers, partners, and even competitors (Meireles et al. 2022). In summary, open innovation recognizes that valuable ideas and technologies are not solely confined within the boundaries of a single organization.

The increasing complexity and pace of technological advancements have made it challenging for any single organization to maintain a monopoly on innovation. As recognized by Delbono and Lambertini (2022), monopolies of innovation can stifle competition, limit consumer choice, and impede overall progress in various industries. Open innovation, on the other hand, promotes a more inclusive and participatory approach to innovation. One of the primary ways open innovation combats monopolies is by breaking down barriers to entry and democratizing access to resources and expertise. By encouraging collaboration between companies, research institutions, startups, and even individuals, open innovation enables a diverse range of actors to contribute ideas, skills, and resources to the innovation process. This democratization of innovation not only fosters competition but also encourages a wider distribution of the benefits of technological progress (Bogers et al. 2018). Moreover, open innovation facilitates the exchange of knowledge and ideas across organizational boundaries. By sharing insights, best practices, and even intellectual property, participants in open innovation ecosystems can collectively overcome challenges and accelerate the pace of innovation. This collaborative approach not only reduces duplication of effort but also enables participants to leverage each other's strengths and capabilities (Pedersen et al. 2022).

The rise of digital technologies and the internet has facilitated greater connectivity and collaboration among individuals and organizations worldwide. Online platforms and tools enable organizations to connect with external partners, including customers, suppliers, and even competitors, to cocreate solutions. Platforms like crowdsourcing websites, innovation marketplaces, and open-source communities provide avenues for diverse stakeholders to contribute ideas, expertise, and resources to innovation processes (Cano et al. 2022; Vignieri 2021). Schlagwein et al. (2017) described the digital technologies that support open innovation through open standards and application programming interfaces (APIs), allowing interoperability and integration between different systems and platforms. By engaging a broad range of industry players, academia, government agencies, and user communities, open innovation initiatives ensure that standards reflect the needs and perspectives of all relevant parties. This inclusivity helps to build consensus around standards, increasing their legitimacy and adoption across industries (Pilena et al. 2021).

Researchers such as Osorno-Hinojosa et al. (2022) and Portuguez-Castro (2023) have indicated that open innovation facilitates cocreation by breaking down traditional barriers between organizations, allowing for the pooling of diverse perspectives and knowledge. It is also recognized that open innovation facilitates cocreation by expanding the innovation ecosystem. By involving a broader range of stakeholders, including customers, suppliers, academia, and even competitors, organizations can access a rich diversity of ideas and insights. This inclusivity fosters creativity and generates novel solutions that may not have emerged within the confines of a single organization. Moreover, open innovation encourages transparent communication and knowledge sharing. By openly sharing information and resources, organizations can build trust and collaboration among participants. This exchange of ideas stimulates iterative feedback loops, enabling continuous improvement and the refinement of solutions through collective effort (Adamides et al. 2023). Additionally, open innovation promotes agility and adaptability. By tapping into external networks, organizations can quickly identify emerging trends, market demands, and technological advancements. This agility allows for rapid iteration and adjustment, ensuring that cocre-

ated solutions remain relevant and competitive in a dynamic environment (Almeida 2021; Andriyani et al. 2024).

Open innovation also promotes the exchange of information about research, development, and best practices. By leveraging external expertise, organizations gain access to new perspectives and insights, leading to more informed decision-making processes. This exchange of knowledge reduces the likelihood of duplicating efforts and encourages the dissemination of valuable information across industries (Weissenberger-Eibl and Hampel 2021). Chiu and Lin (2022) added that open innovation enhances the transparency of product development and supply chains. By involving stakeholders at various stages of the innovation lifecycle, organizations can address potential issues early on and respond to changing market demands more effectively. This collaborative approach also enables greater traceability and accountability, as participants have insight into the origins and processes involved in creating products and services.

For SMEs, open innovation offers several advantages. Firstly, it allows them to access a broader pool of expertise and resources that they may not possess internally. SMEs often have limited R&D budgets and human resources, making it challenging to compete with larger firms in terms of innovation. By tapping into external networks, SMEs can gain access to specialized skills, technologies, and funding opportunities that can accelerate their innovation process (Annamalah et al. 2022). It was also advocated by Farjam et al. (2023) that open innovation enables SMEs to mitigate the risks associated with innovation. Collaborating with external partners allows them to share the financial burden of R&D investments and reduces the likelihood of failure. Additionally, by involving customers and other stakeholders in the innovation process, SMEs can ensure that their products or services meet market needs and are more likely to be adopted.

Recent studies like Bekata and Kero (2024) and Ta'Amnha et al. (2023) have suggested that open innovation cultivates an entrepreneurial mindset within SMEs by promoting collaboration and networking with external partners such as startups, research institutions, and other businesses. Through these partnerships, SMEs can tap into new markets, technologies, and business models that they might not have explored otherwise. This exposure to external ecosystems nurtures an entrepreneurial spirit within the organization, fostering a culture of risk taking, experimentation, and agility. Moreover, open innovation encourages SMEs to be more flexible and adaptive to changes in the business environment. By continuously scanning the external landscape for emerging trends and opportunities, SMEs can stay ahead of the curve and seize new growth avenues. This proactive approach to innovation instills a sense of empowerment and ownership among employees, inspiring them to contribute ideas and take initiative in driving the company forward. Employees can actively participate in idea-generation sessions, brainstorming meetings, and innovation workshops to share their insights and contribute to the development of new products, services, or processes (Gama et al. 2019). Encouraging a mindset of experimentation and risk taking enables employees to explore unconventional solutions and challenge the status quo.

## 3. Hypothesis Development

As a theoretical lens, this study followed the work carried out by Cricelli et al. (2023), who provided a comprehensive framework to identify the causes of failure of open innovation. Seven main dimensions of open innovation failures were identified, as reported in Table 1. The causes reported in each dimension and the frequency of their occurrence were identified, which corresponded to the number of themes identified in the systematic review reported by Cricelli et al. (2023). The causes of failure in open innovation were not all equally prominent. For instance, knowledge and IP management was the most frequent dimension ($n$ = 111), while strategy-related issues dimension was only reported in 46 studies. Martinez-Conesa et al. (2017) also indicated that SMEs often lack the resources and expertise to effectively manage the flow of knowledge both within and outside their organization, which can lead to difficulties in identifying valuable external knowledge sources and integrating them into their innovation processes. In this sense, it was important

to explore a first research hypothesis that aimed to determine whether the relevance of these dimensions is also the same in the SME segment. Accordingly, the first hypothesis was established:

**H1.** *The dimensions of the causes of failure in open innovation are different for SMEs.*

**Table 1.** Causes of open innovation failures (adapted from Cricelli et al. 2023).

| Dimension | Frequency | No. of Causes | Cause |
|---|---|---|---|
| Strategy-related | 56 | 4 | Misalignment between partners' goals<br>Prevalence of closed innovation model<br>Lack of dynamic capabilities<br>Lack of an adequate business model |
| Organizational structure | 74 | 4 | Inadequate coordination and communication mechanisms<br>Inadequate reward and control systems<br>Misalignment between partners' organizational structure<br>Rigid organizational structure/excessive bureaucracy |
| Organizational culture | 70 | 3 | Not invented here/not sold here syndromes<br>Misalignment between partners' organizational cultures<br>Individual level resistances |
| Knowledge and IP management | 111 | 3 | Loss of know-how/competitive advantage<br>Inadequate appropriability systems<br>Lack of absorptive/desorptive capacity |
| Management's skills and actions | 76 | 3 | Lack of experience in OI management<br>Incorrect cost–benefit assessment<br>Ineffective scan of environment |
| Resources | 88 | 4 | Inadequate technology/ICT<br>Inadequate IPs and asset<br>Management<br>Inadequate HR management<br>Lack of economic/financial resources |
| Interfirm collaboration | 96 | 3 | Opportunistic behavior/free<br>Riding<br>High transaction costs<br>Lack of trust |

Furthermore, it is important to deepen our knowledge of each dimension and analyze the relevance of the specific causes that make up these dimensions for SMEs. Exploring these factors is relevant because an SME possesses several distinctions from larger corporations.

SMEs and large companies differ significantly in their approach to strategy due to their size, resources, and organizational structure. SMEs often exhibit flexibility and agility in their strategy formulation and execution, leveraging their ability to adapt quickly to market changes (Arsawan et al. 2022; Puriwat and Tripopsakul 2021). Furthermore, organizational agility and open innovation are symbiotic forces driving competitiveness in today's dynamic business landscape. Organizational agility fosters adaptability, allowing firms to swiftly respond to market shifts, technological advancements, and customer needs. This flexibility enables firms to embrace open innovation practices, collaborating with external partners, such as startups, academia, or customers, to co-create value (Zhang et al. 2023). Accordingly, they tend to focus on niche markets or specialized products/services, aiming for differentiation to compete effectively.

**H2.** *The strategy-related dimension is different for SMEs.*

Organizational structure and culture vary significantly between SMEs and large companies. SMEs typically have a flatter organizational structure, with fewer hierarchical levels and a more informal communication flow. As pointed out by Kindström et al. (2022), decision making in SMEs tends to be decentralized, allowing for quick responses to market changes and employee empowerment. Hassi et al. (2022) found that empowering

employees involves granting them the autonomy and authority to make choices relevant to their roles. In contrast, large companies often have complex hierarchical structures with multiple layers of management, leading to slower decision-making processes and greater bureaucracy.

**H3.** *The organizational structure dimension is different for SMEs.*

**H4.** *The organizational culture dimension is different for SMEs.*

Knowledge management and IP processes are also another factor to explore. SMEs often rely heavily on tacit knowledge, which resides in the minds of employees and is informal in nature. Cerchione et al. (2015) stated that knowledge sharing in SMEs tends to be organic, happening through interpersonal communication and experiential learning. However, formalized knowledge management systems may be lacking due to resource constraints. In contrast, large companies typically have structured knowledge management systems, including databases, intranets, and collaboration tools to capture, store, and disseminate knowledge across the organization. SMEs often operate with leaner teams, where employees need to perform several tasks and possess a diverse set of skills to fulfill various roles. Cross-functional training and on-the-job learning are common in SMEs, fostering a culture of versatility and adaptability (Efstathiades et al. 2016). However, SMEs may struggle with formalized skill development programs due to limited resources, as reported by Deschênes (2023) and Panagiotakopoulos (2011).

**H5.** *The knowledge and IP management dimension is different for SMEs.*

**H6.** *The management's skills and actions dimension is different for SMEs.*

**H7.** *The resources dimension is different for SMEs.*

Finally, SMEs often lack the resources and capabilities to compete effectively on their own in increasingly complex and dynamic markets. Collaborating with other firms allows them to pool resources, share expertise, and access new markets or technologies that may be beyond their individual reach. Furthermore, collaboration enables SMEs to mitigate the risks associated with market uncertainties, economic fluctuations, and rapid technological advancements by diversifying their networks and spreading the burden of innovation and investment (Mthiyane et al. 2022). Additionally, Audretsch et al. (2023) and Castellani et al. (2023) have revealed that partnering with other firms can facilitate knowledge exchange, learning opportunities, and synergistic innovation, fostering a culture of continuous improvement and competitiveness.

**H8.** *The interfirm collaboration dimension is more relevant for SMEs.*

## 4. Materials and Methods

This study adopted a quantitative methodology to quantify and explored the relative relevance of the causes of failure of open innovation among SMEs. Quantitative methods provide a structured framework for data collection and analysis. Furthermore, this approach contributes to identifying patterns and correlations among different variables. Quantitative approaches also facilitate the identification of causal relationships, helping to pinpoint specific factors that significantly impact the success or failure of open innovation initiatives. This can guide organizations in focusing their efforts on addressing critical issues and implementing targeted interventions to improve future outcomes.

This study also aimed to explore the prevalence of these causes among SMEs in contrast to the findings identified by Cricelli et al. (2023), in which the causes for identifying failures in open innovation resulted from secondary sources mainly made up of large organizations.

As such, this study applied hypothesis testing for the difference between two means, as indicated in expression (1),

$$t = \frac{(\overline{X1} - \overline{X2})}{\sqrt{\frac{S1^2}{n1} + \frac{S2^2}{n2}}}$$

(1)

where $\overline{X1}$ and $\overline{X2}$ are the means of the two samples, $S1$ and $S2$ are the standard deviations, and $n1$ and $n2$ are the sample sizes. It is a statistical method used to determine whether there is a significant difference between the means of two populations or groups. Jean (2017) stated that hypothesis testing for the difference of two means provides a systematic and objective approach for making inferences about population parameters based on sample data, helping researchers draw conclusions about the effectiveness of interventions or the presence of differences between groups. Data were explored and analyzed using the IBM SPSS software v.27.

Data were collected from SMEs registered in Portugal that implemented open innovation practices with activity reported for 2021. Open innovation could have been practiced at both the level of the entire businesses and as individual projects within those businesses. In an open innovation business, the entire organization adopts principles and practices of open innovation across all departments, functions, and activities. This means that the company actively seeks and incorporates external ideas, technologies, and partnerships into its overall strategy and operations. Open innovation projects, on the other hand, refer to specific initiatives or endeavors within a company where open innovation principles are applied. These projects may involve collaboration with external partners, sharing of knowledge and resources, and leveraging external expertise to achieve specific goals or develop particular products/services. An online survey was created and disseminated between October and December 2022. Two response reminders were sent out at the end of November and at the end of the second week of December. A total of 366 responses were received, but only 297 responses were considered valid after eliminating null and duplicate responses from the same company. Table 2 presents the territorial distribution of the sample in Portugal adopting the NUTS II framework. Information about the absolute frequency (AF), relative frequency (RF), cumulative absolute frequency (CAF), and cumulative relative frequency (CRF) is given. NUTS is a hierarchical classification system developed by the European Union for dividing the territory of its member states and other countries into regions. It provides a standardized framework for collecting and reporting statistical data across different administrative levels, facilitating comparability and harmonization of statistical information for various purposes such as economic analysis, policy planning, and regional development. NUTS divides territories into three hierarchical levels: NUTS 1 regions (major socioeconomic regions), NUTS 2 regions (basic regions for the application of regional policies), and NUTS 3 regions (smaller administrative units). The InformaDB database was used to extract information regarding the location and activity of each company. InformaDB is a comprehensive database designed to provide detailed information on SMEs. It serves as a repository of data encompassing various aspects of SMEs, including their financial performance, market dynamics, operational metrics, and industry-specific insights. For studies concerning SMEs, InformaDB holds significant importance due to several reasons like its relevance for economic development, employment generation, and innovation (Curado et al. 2022; Picas et al. 2021).

**Table 2.** Sample characteristics.

| NUTS II | AF | RF | CAF | CRF |
|---|---|---|---|---|
| North | 93 | 0.3131 | 93 | 0.3131 |
| Algarve | 21 | 0.0707 | 114 | 0.3838 |
| Center | 56 | 0.1886 | 170 | 0.5724 |
| Setúbal Peninsula | 18 | 0.0606 | 188 | 0.6330 |
| Lisbon Metropolitan Area | 61 | 0.2054 | 249 | 0.8384 |
| Alentejo | 10 | 0.0337 | 259 | 0.8721 |

**Table 2.** *Cont.*

| NUTS II | AF | RF | CAF | CRF |
|---|---|---|---|---|
| West and the Tagus Valley | 21 | 0.0707 | 280 | 0.9428 |
| Azores | 4 | 0.0135 | 284 | 0.9562 |
| Madeira | 13 | 0.0438 | 297 | 1 |

## 5. Results and Discussion

This study first explored whether were differences in the causes of failure between the data obtained from the systematic review carried out by Cricelli et al. (2023) and the data obtained in this study from the survey of SMEs. It was necessary to standardize the scales to [0–100] to ensure the data from both studies were comparable, and two new metrics were obtained: the relative weight of the dimension in the systematic review (SLR-CF) and the relative weight of that dimension in the SME survey (SME-CF). The findings shown in Figure 1 answered H1 and showed that

- The resources dimension was the main cause of failure of SMEs, while the systematic review identified the knowledge and IP management dimension as the main cause.
- In SMEs, the organizational structure and organizational culture dimensions were less important. They were the two least important causes.
- The management skill and actions dimension was more relevant for SMEs, emerging as the second most important cause of failure.
- The relative importance of the causes related to strategy, intellectual property management, and interfirm collaboration were identical in both studies.

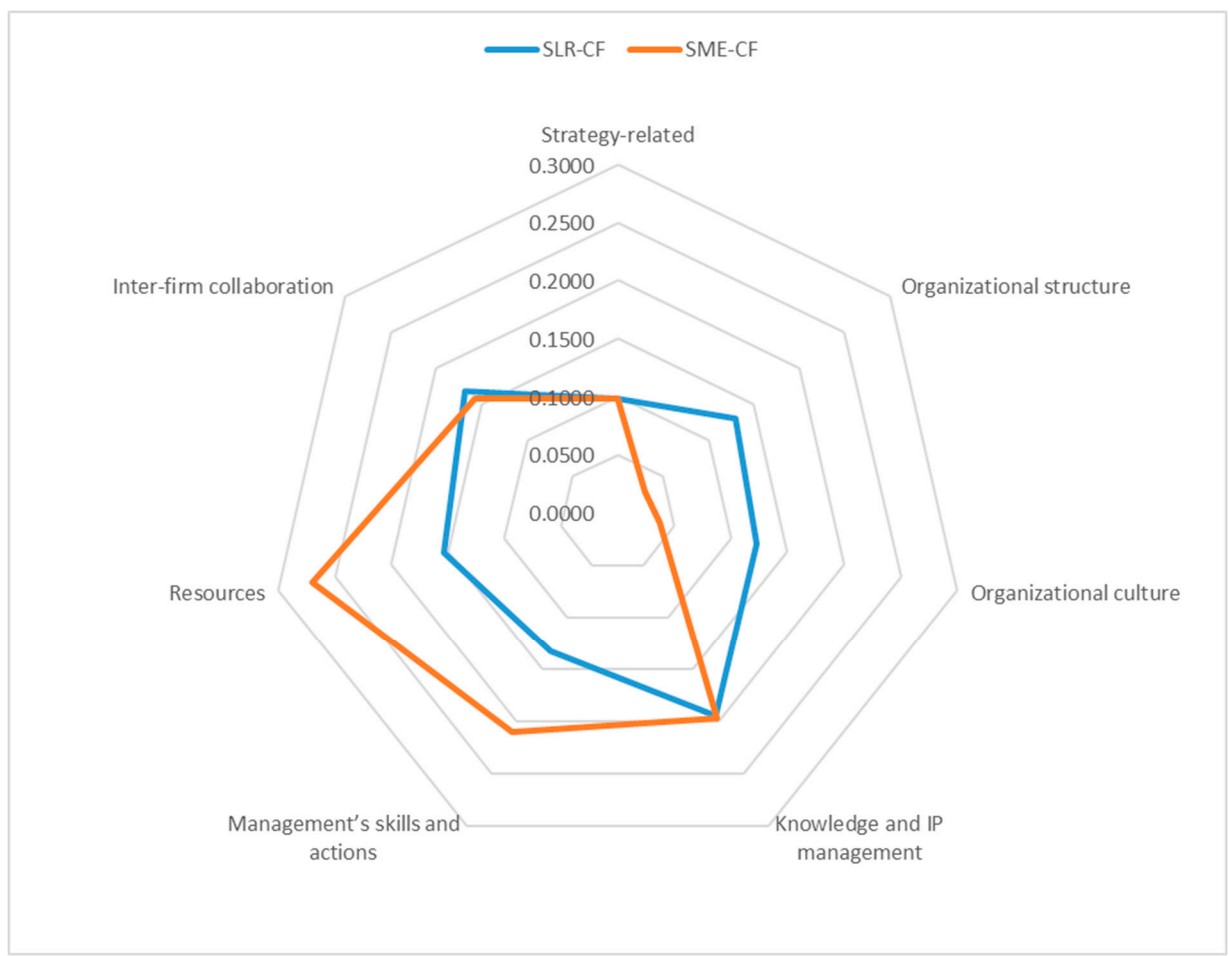

**Figure 1.** Comparative analysis of the causes of failure in open innovation.

The results of this study are aligned with the findings reported by Dubouloz et al. (2021) and Kraus et al. (2020), which indicated that SMEs face numerous challenges when it comes to engaging in open innovation due to their limited resources and capabilities. These challenges include financial constraints, lack of human capital, limited access to external networks, and intellectual property concerns. However, this last point is not confirmed by this study, as the same difficulties in managing intellectual property were reported in both SMEs and large organizations. SMEs can mitigate this issue by seeking legal advice from IP professionals to navigate complex IP issues and mitigate risks effectively. The results also indicated that SMEs were adopting effective strategies to reduce the organizational structure and organizational culture to perform open innovation. They could implement a more flexible and decentralized organizational structure that encourages cross-functional collaboration and facilitates communication across different departments or teams. This structure allows for greater agility and responsiveness to external opportunities and challenges, enabling SMEs to engage in open innovation initiatives more effectively. Pierre and Fernandez (2018) suggested the creation of dedicated innovation units or teams within their organization to focus specifically on exploring and pursuing open innovation opportunities. These teams can be tasked with scouting for external partners, monitoring industry trends, and managing collaborative projects, thereby streamlining the innovation process, and ensuring that it receives the necessary attention and resources. In terms of organizational culture, SMEs can cultivate a more open and inclusive environment that values creativity, experimentation, and risk taking. This involves fostering a culture of trust and transparency where employees feel empowered to share ideas, challenge the status quo, and collaborate with external partners. It emerged that implementing idea sharing and collaboration in SMEs was often easier than in larger organizations due to several key factors. Firstly, SMEs typically have fewer layers of hierarchy and less bureaucratic red tape, which facilitate faster decision making and communication. This agility allows ideas to be shared more freely and acted upon promptly, without being bogged down by extensive approval processes or departmental silos. Secondly, the smaller size of SMEs fosters a more close-knit and cohesive work environment. Employees often have direct access to management and feel more comfortable voicing their opinions and contributing ideas. This accessibility encourages a culture of openness and innovation, as recognized by Rumanti et al. (2023), where everyone's input is valued and considered.

The next step aimed to explore the specific behavior regarding the causes for each dimension. The following hypothesis test was carried out: μSME-CF-μSLR-CF ≠ 0, corresponding to each dimension, and the *p*-value was determined considering a significance level of 0.05. Table 3 shows the results obtained for the strategy-related dimension. There is no difference in behavior between the two studies for the four causes of this dimension. Accordingly, H2 was rejected.

**Table 3.** Differences in strategy-related dimension.

| Cause | AF (SLR-CF) | RF (SLR-CF) | AF (SME-CF) | RF (SME-CF) | *p*-Value |
|---|---|---|---|---|---|
| Misalignment between partners' goals | 18 | 0.3214 | 23 | 0.2987 | 0.3906 |
| Prevalence of closed innovation model | 17 | 0.3036 | 20 | 0.2597 | 0.0983 |
| Lack of dynamic capabilities | 12 | 0.2143 | 19 | 0.2468 | 0.2198 |
| Lack of an adequate business model | 9 | 0.1607 | 15 | 0.1948 | 0.1980 |

Table 4 explores the organizational structure dimension. It was concluded that H3 could be accepted. For large companies, inadequate coordination and communication mechanisms were identified as the main cause of failure, while more than 60 percent

of SMEs identified inadequate reward and control systems. Only the cause related to misalignment between partners' organizational structure was similar in both studies. A common issue in SMEs is the lack of clear reward structures for employees who contribute to innovation efforts. Unlike larger corporations with established innovation departments and incentive programs, SMEs often lack formal mechanisms to recognize and reward innovative ideas or efforts. Furthermore, Khan et al. (2021) pointed out that the limited resources available to SMEs can exacerbate these challenges. Many SMEs operate on tight budgets, making it difficult to invest in the infrastructure and expertise needed to establish robust reward and control systems for innovation. This study confirms the findings obtained by Spithoven et al. (2013) in that the rigid organizational structure and excessive bureaucracy found in large companies can act as significant barriers to innovation, whereas SMEs, with their more flexible and dynamic environments, are better positioned to foster innovation and adaptability.

**Table 4.** Differences in organizational structure dimension.

| Cause | AF (SLR-CF) | RF (SLR-CF) | AF (SME-CF) | RF (SME-CF) | *p*-Value |
|---|---|---|---|---|---|
| Inadequate coordination and communication mechanisms | 22 | 0.2973 | 2 | 0.0870 | $<1 \times 10^{-3}$ |
| Inadequate reward and control systems | 18 | 0.2432 | 14 | 0.6087 | $<1 \times 10^{-3}$ |
| Misalignment between partners' organizational structure | 17 | 0.2297 | 6 | 0.2609 | 0.3893 |
| Rigid organizational structure/excessive bureaucracy | 17 | 0.2297 | 1 | 0.0435 | $<1 \times 10^{-3}$ |

Table 5 shows the differences in the organizational culture dimension. The findings support the acceptance of H4. SMEs face more not invented here (NIH)/not sold here (NSH) syndromes but exhibit less individual-level resistance. According to Amann et al. (2022), the NIH syndrome reflects a reluctance to adopt or embrace innovations, technologies, or ideas created externally, often stemming from a belief that internally generated solutions are superior. Conversely, the NSH syndrome refers to the hesitance to commercialize or adopt innovations that originated elsewhere. Both the NIH and NSH syndromes can impede the success of open innovation initiatives by fostering insularity and limiting the exchange of knowledge and resources between organizations. This study also indicates that SMEs often encountered less individual-level resistance in open innovation. Several reasons can justify this behavior. Firstly, SMEs typically possess a flatter organizational structure than larger corporations. This allows for more direct communication and collaboration among team members, fostering a culture of openness and receptivity to new ideas. In such environments, employees are often more willing to embrace change and experimentation, reducing resistance to new innovation initiatives. Moreover, SMEs often have a more cohesive and closely knit workforce. This sense of unity and shared purpose can mitigate individual resistance to innovation, as employees feel a greater sense of ownership and involvement in the company's success. As reported by Alkhalaf and Al-Tabbaa (2024), employees in SMEs may have greater autonomy and flexibility in their roles, empowering them to contribute ideas and participate in innovation efforts more freely.

**Table 5.** Differences in organizational culture dimension.

| Cause | AF (SLR-CF) | RF (SLR-CF) | AF (SME-CF) | RF (SME-CF) | *p*-Value |
|---|---|---|---|---|---|
| Not invented here/not sold here syndromes | 30 | 0.4286 | 17 | 0.5862 | $<1 \times 10^{-3}$ |
| Misalignment between partners' organizational cultures | 22 | 0.3143 | 10 | 0.3448 | 0.3614 |
| Individual level resistances | 18 | 0.2571 | 2 | 0.0690 | $<1 \times 10^{-3}$ |

Table 6 shows the causes of failure for knowledge and IP management dimension. The findings indicated that H5 could be rejected. The causes of failure were equally important regardless of the profile of the organization.

**Table 6.** Differences in knowledge and IP management dimension.

| Cause | AF (SLR-CF) | RF (SLR-CF) | AF (SME-CF) | RF (SME-CF) | *p*-Value |
|---|---|---|---|---|---|
| Loss of know-how/competitive advantage | 48 | 0.4324 | 68 | 0.4416 | 0.6228 |
| Inadequate appropriability systems | 42 | 0.3784 | 61 | 0.3961 | 0.3442 |
| Lack of absorptive/desorptive capacity | 21 | 0.1892 | 25 | 0.1623 | 0.1511 |

Table 7 explores the relevance of the causes for the results regarding the management skills and actions dimension. It was concluded that H6 could be accepted. All causes of failure presented *p*-values below 0.05. In particular, SMEs often encountered challenges in managing open innovation due to their limited resources and experience. One significant obstacle reported by Odriozola-Fernández et al. (2019) was the lack of familiarity with the intricacies of open innovation frameworks and practices. Unlike larger corporations with dedicated departments and experienced professionals, SMEs may lack the necessary expertise in identifying, nurturing, and implementing open innovation initiatives effectively.

**Table 7.** Differences in management skills and actions dimension.

| Cause | AF (SLR-CF) | RF (SLR-CF) | AF (SME-CF) | RF (SME-CF) | *p*-Value |
|---|---|---|---|---|---|
| Lack of experience in OI management | 35 | 0.4605 | 95 | 0.5793 | $<1 \times 10^{-3}$ |
| Incorrect cost–benefit assessment | 22 | 0.2895 | 36 | 0.2195 | 0.0010 |
| Ineffective scan of environment | 19 | 0.2500 | 33 | 0.2012 | 0.0204 |

Table 8 explores the differences in the resource dimension. Only failure relating to inadequate HR management was not significant. H7 was therefore accepted. Lack of economic/financial resources and inadequate IP and asset management were the two causes where there were the greatest differences between the organizations. SMEs may lack the internal expertise and infrastructure required to effectively manage open innovation processes. Implementing collaborative platforms, establishing partnerships, and managing intellectual property rights entail considerable costs and expertise. SMEs may find it

challenging to attract and retain skilled personnel capable of driving innovation initiatives forward. Another reason was reported by Lewandowska (2009), related to limited access to capital. Unlike large corporations, SMEs typically have tighter budgets and may struggle to allocate funds toward the research and development (R&D) initiatives necessary for open innovation. Securing external funding can be difficult, as investors may perceive SMEs as riskier ventures than larger, more established companies. Furthermore, external asset management may allow SMEs to optimize their resource allocation and focus on their core competencies. By outsourcing certain functions or assets to specialized providers, such as intellectual property management firms or patent attorneys, SMEs can streamline operations and allocate their limited resources more effectively (Seepana et al. 2022). This strategic approach enables SMEs to stay agile and competitive in dynamic market environments. The findings also revealed the importance of SMEs using technology to enhance their agility and competitiveness. Investing in digital tools and platforms can streamline operations, improve efficiency, and facilitate faster decision-making processes (Almeida and Wasim 2023; Hautala-Kankaanpää 2023). For example, adopting cloud-based solutions for data management and collaboration enables SMEs to scale their operations rapidly and access real-time insights for informed decision making (Neicu et al. 2020).

**Table 8.** Differences in resources dimension.

| Cause | AF (SLR-CF) | RF (SLR-CF) | AF (SME-CF) | RF (SME-CF) | *p*-Value |
|---|---|---|---|---|---|
| Inadequate technology/ICT | 22 | 0.2500 | 41 | 0.1943 | 0.0039 |
| Inadequate IPs and asset management | 30 | 0.3409 | 46 | 0.2180 | $<1 \times 10^{-3}$ |
| Inadequate HR management | 24 | 0.2727 | 50 | 0.2370 | 0.0625 |
| Lack of economic/financial resources | 12 | 0.1364 | 74 | 0.3507 | $<1 \times 10^{-3}$ |

Finally, Table 9 presents the results for the interfirm collaboration dimension. None of the causes of failure had *p*-values lower than 0.05, and therefore H8 was rejected. It was concluded that the behavior of organizations, regardless of their size, is identical when it comes to establishing interfirm collaboration.

**Table 9.** Differences in interfirm collaboration dimension.

| Cause | AF (SLR-CF) | RF (SLR-CF) | AF (SME-CF) | RF (SME-CF) | *p*-Value |
|---|---|---|---|---|---|
| Opportunistic behavior/free riding | 39 | 0.4063 | 53 | 0.4309 | 0.2299 |
| High transaction costs | 33 | 0.3438 | 40 | 0.3252 | 0.5774 |
| Lack of trust | 24 | 0.2500 | 30 | 0.2439 | 0.7655 |

## 6. Conclusions

Of the eight hypotheses formulated, five were accepted, which indicated that the causes of open innovation failure were significantly different for large companies and SMEs. Firstly, the dimensions of resources and management's skills and actions are the main factors hindering the implementation of open innovation practices among SMEs. SMEs' limited resources can act as an inhibitor to open innovation. SMEs often face financial constraints, which make it difficult to allocate resources to open innovation initiatives, such as partnerships with other companies, universities, or research institutions. Moreover, a lack of knowledge about how to effectively access and use external resources can make it even more difficult to engage in open innovation practices. Furthermore, the lack of specific managerial skills to deal with the complexity of open innovation can represent a

significant challenge for SMEs. Leaders without adequate experience or knowledge of how to identify opportunities for external collaboration, establish strategic partnerships, and manage relationships with other organizations may feel insecure or unable to effectively lead open innovation initiatives, which arises more frequently with SMEs. Despite this, organizational and cultural factors help SMEs in open innovation processes. Agile and flexible organizations are able to adapt quickly to changes in the business environment and exploit innovation opportunities effectively. Flat, decentralized organizational structures can facilitate collaboration and the exchange of ideas between employees, encouraging the generation of new solutions and the implementation of innovative practices. Moreover, a corporate culture that values creativity, continuous learning, and experimentation is key to promoting open innovation. SMEs that encourage the active participation of employees in the generation of ideas and in the innovation process tend to be more successful in implementing open innovation practices.

Exploring the causes of failure in each dimension led to the conclusion that there were differences for SMEs in the dimensions of organizational structure, organizational culture, management's skills and actions, and resources. SMEs had inadequate reward and control systems. Typically, SMEs may not have the financial resources to invest in robust reward programs, which can result in inadequate incentives for employees to actively participate in innovation. In addition, SMEs may have less-formalized organizational structures, which makes it more difficult to establish effective control systems to monitor and evaluate the progress of open innovation. In addition, due to their limited resources, SMEs may not have the capacity to dedicate significant time and resources to exploring and understanding the concepts and practices of open innovation. This can result in a lack of exposure and familiarity with the business models and strategies associated with open innovation. Finally, this study also found that SMEs may have leaner organizational structures and fewer specialists dedicated to innovation management than large companies. This implies that they may lack the specialized skills needed to effectively develop and implement open innovation initiatives, such as managing external partnerships, open collaboration, and intellectual property management.

This study offers significant theoretical and practical implications that need to be addressed. This work theoretically enriches the field of open innovation by offering valuable insights into the underlying mechanisms that lead to the failure of implementing this approach, particularly among SMEs. In this sense, this study helps to fill the gaps in existing knowledge by providing a deeper understanding of the complexities involved in managing open innovation, particularly through a more in-depth analysis of the factors that influence its success or failure. By studying the reasons for the failures of open innovation, this study generates discussion on important conceptual issues, such as the precise definition of open innovation, its limits, and boundaries, as well as the different contexts in which it can be successfully applied. This can lead to the refinement of concepts and theories related to open innovation, promoting a more holistic and accurate understanding of this approach. Furthermore, by identifying patterns and trends in open innovation failures, this study can help develop more robust explanatory models that describe open innovation processes more accurately. These improved theoretical models can, in turn, inform managerial practice and guide strategic decision making in organizations seeking to implement open innovation effectively. From a practical perspective, this study provides valuable guidance for organizations looking to adopt or improve their open innovation practices. It can help companies to identify and better understand the common obstacles that can arise when adopting this collaborative innovation model. This allows them to anticipate possible problems and develop proactive strategies to deal with them. Furthermore, by examining the failures of open innovation, organizations can learn from the mistakes of other companies and avoid repeating the same mistakes. By identifying specific open innovation barriers relevant to their organization, managers can develop targeted strategies to overcome them. For instance, fostering a culture that values collaboration and risk taking can help mitigate resistance to sharing ideas externally. Establishing clear IP policies and agreements can

alleviate concerns about IP protection. Moreover, investing in communication channels and platforms to facilitate collaboration with external partners can address coordination challenges. Additionally, this study underscores the need for flexibility and adaptation in open innovation approaches. Managers should be prepared to iterate and refine their strategies based on ongoing assessments of barriers and feedback from stakeholders. This practical contribution is especially relevant for SMEs that have fewer resources, are less mature, and experience greater difficulties in managing open innovation.

This study has some limitations that are worth considering. Firstly, the specific characteristics of the SMEs in the sample were not explored. In fact, SMEs can be very heterogeneous in terms of their size, sector of activity, and age, which can lead to specific results for these segments. In this sense, it is suggested that future work in the area could explore the role and impact of these factors. It is also important to explore SMEs' previous experience with open innovation processes. The process of becoming involved in open innovation is expected to be a process of continuous learning. It is also recommended that future work in this area explore the role of SMEs' relationships with regional and local business structures and with scientific and academic entities such as universities and research centers. SMEs are also heterogeneous in terms of their geographical location. In this sense, it is suggested that this study be replicated in other contexts inside and outside Europe. It would be interesting to explore whether the causes of failure in open innovation are similar in other contexts. Finally, this study did not make any distinction regarding the scope of implementing open innovation. Future studies should consider two distinct samples composed by companies that adopt open innovation businesses and other with companies that participate in open innovation projects or initiatives.

**Funding:** This research received no external funding.

**Institutional Review Board Statement:** Not applicable.

**Informed Consent Statement:** At the beginning of each survey, it was clearly stated that the response data obtained through this survey would only be used for educational and research purposes. The survey was conducted multiple times, and SMEs were coded specifically for matching purposes but were not individually identifiable.

**Data Availability Statement:** The data used in this study are available from the corresponding author upon reasonable request.

**Conflicts of Interest:** The author declares no conflicts of interest.

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
