# Peer review of "Causes of Failure of Open Innovation Practices in Small- and Medium-Sized Enterprises"

_admsci, doi:10.3390/admsci14030050_

Round 1

Reviewer 1 Report

Comments and Suggestions for Authors

Quite interesting study about failures of open innovation failures in SMEs. Data collected in a sample of Portugues SMEs.

The hypothesis preparation is very good and relevant to the open innovation theory.

The authors should refer to the book Managing open innovation in SMEs by W. Wanhaverbeke. I also appreciate the practical guide 'A comprehensive guide to “Efficient Open Innovation”' which is not scientific source but I know practictioners that subscribe this point of view. 

The main remark to the authors: in the paper is not explained that we consider open innovation businesses or projects. Typical open innovation ventures or project are very seldom and they are undertaken by large companies. My concern is that respondents commented on certain open features of all innovations they have encountered. Therefore, I suggest that the authors should:

1. Explain how the respondents were selected in terms of the openness of the innovations they use.

2. In limitations describe the problem of understanding open innovation by the respondents. I am afaid we can the authors can make validity error.

Author Response

Thank you very much for your improvement suggestions. Pleas find attached my responses.

Reviewer 2 Report

Comments and Suggestions for Authors

Dear authors, thank you very much for a nice work. I am so happy to be the first reader of your paper. Please find my comments below and i do hope you will find them useful to improve readability of your paper, as well as contribution to the literature.

With regards to hypothesis. I believe it can be improved by introducing references from authors who studied barriers to innovation. There is little justification for the barriers,  please put them in literature by one by and before explain why you propose such hypotheses.

I believe authors can improve the conclusion section by suggestion practical implication especially for managers.

All items i mentioned are major issues and please to your best to revize. And  i believe you will do it.

Good luck.

Comments on the Quality of English Language

Quality of communication is good

Author Response

(The authors gave the same response as above.)
